# *Akkermansia muciniphila,* a New Generation of Beneficial Microbiota in Modulating Obesity: A Systematic Review

**DOI:** 10.3390/microorganisms9051098

**Published:** 2021-05-20

**Authors:** Jumana Nabil Abuqwider, Gianluigi Mauriello, Mohammad Altamimi

**Affiliations:** 1Department of Nutrition and Food Technology, Faculty of Agriculture and Veterinary Medicine, An-Najah National University, Nablus P.O. Box 7, Palestine; jqwider@gmail.com (J.N.A.); m.altamimi@najah.edu (M.A.); 2Department of Agricultural Science, University of Naples Federico II, 80049 Naples, Italy

**Keywords:** new generation probiotic, glucose homeostasis, lipid profile

## Abstract

Obesity is a complex syndrome and is recognized as the ultimate pathway of many chronic diseases. Studies using *Akkermansia muciniphila* supplementation strategy have proved to be effective for the prevention and treatment of obesity and other metabolic disorders. Although there are studies that support the protective effect of this strategy, the effects on the prevention of obesity on humans are not clear yet and need more investigation. The aim of this study is to investigate the effect of *A. muciniphila* administration on modulating obesity. This systematic review was generated from articles published within the last 10 years. All articles were in English and included animal subjects. The review relied on the search engines Google Scholar, Pub Med, Web of Science and Medline using the following keywords: *A. muciniphila*, next-generation probiotic, new-generation probiotic, obesity, fat mass, body fat and lipid profile. The search has revealed 804 articles with relevant key words. After the exclusion of irrelevant articles, 10 studies were selected based on the criteria. These studies were randomized controlled trials that have shown that *A. muciniphila* modulates obesity by regulating metabolism and energy hemostasis and improving insulin sensitivity and glucose hemostasis. In addition, studies showed this microorganism enhances low grade inflammation by different mechanisms.

## 1. Introduction

Disparity in energy consumption and energy expenditure is manifested as weight gain and adiposity [1]. Obesity has been recognized as one of the most severe public-health issues in the 21st century due to its detrimental impact on overall quality of life [2]. Obesity is associated with the prevalence of many chronic diseases, such as cardiovascular disorders, certain tumors, type 2 diabetes, elevated blood pressure, stroke, osteoarthritis, gallbladder disease, and psychosocial issues [3]. Overweight and obesity were estimated to cause 3.4 million deaths globally in 2010 [4], and have affected more than 671 million in 2016 [5]. Obesity has recently been shown to be a progressive phenomenon in both developed and developing countries [6], and the prevalence of obesity has tripled between 1975 and 2016 according to the WHO [7]. Rural areas have the highest increase compared to cities, meaning that dramatic changes have happened in the lifestyle [8]. The main culprit of such changes was overconsumption of animal products, refined grains and added sugars, especially in forms of sweetened beverages [9]. On the other hand, macronutrients including carbohydrates and proteins have been demonstrated to be the most influential in restructuring the gut microbiota [10]. Simply put, the variations in the gut microbiota rely on the variations presented in the carbohydrates (i.e., sugars) [10].

In both human and animal studies, gut microbes have been reported to play a pivotal role in regulating host metabolism [11]. Tens of trillions of microorganisms and more than 1000 different species of bacteria, with at least 3 million genes, have been identified in the microbiota of the human intestine. The classification of species among members of the gut community is enormous because the distribution and composition of gut microbiota differs at different anatomical sites of the intestine and is likely to be affected by intrinsic and extrinsic factors, including lifestyle, diet, and health status [12]. The colon was reported to be the most inhabited site, with approximately 10^7^ to 10^8^ cells of *Clostridium* Type IV and XIV, *Bacteroidetes*, *Bifidobacterium* and *Enterobacteriaceae* [13]. *Firmicutes* and *Bacteroidetes* represent 64% and 23%, respectively, of the microbiota.

Not only the gut microbiota composition has been reported to exhibit strong pathological and physiological associations with obesity and metabolic syndrome [14], but also the ratio of Firmicutes to Bacteroides was found to be crucial [15]. Accordingly, the ratio of Firmicutes to Bacteroides in adults was found to increase as the BMI increased [16]. However, in conflict with these results, several studies did not observe any modifications of the Firmicutes/Bacteroidetes ratio, or even reported a decreased value in obese animals and humans [17].

Mucin, a protective barrier against xenobiotics in the intestine, plays a great role in the microbiota adhesion to the intestinal layers. The bacteria that have the ability to degrade mucin are more advantageous to survive the changing microenvironment of the intestine [18]. In this term, probiotics, to be effective, should adhere to and interact with mucin layers, while pathogenicity of harmful microbes would increase if they were mucin-degrading microorganisms [19].

*Akkermansia muciniphila* is one of the early occupants (in first year of life) of the intestinal tract with 10^8^ cell/gm [19] or more than 1% of total faecal microbes [20]. *Akkermansia muciniphila* can use mucin as its sole source of carbon and nitrogen. It is considered one of the “next-generation probiotics”, and the knowledge about it has grown exponentially since its first isolation in 2004 [20]. Since it is the only member of *Verrucomicrobia* (phylum) in the gut of mammals, it is easier to be detected using 16S rRNA gene sequence. A large number of studies has shown that the abundance of *Akkermansia* in the gut was correlated with host health and disease status [21] and its number as a beneficial microorganism is decreasing by aging [19]. As a matter of fact, one study showed that *A. muciniphila* has modulated the endocannabinoid (eCB) system. In the context of obesity, type 2 diabetes and inflammation, eCB is an essential regulatory system reported to include glucose and energy metabolism [22]. Human and animal trials showed a positive correlation between *A. muciniphila* intervention and obesity and metabolic disorders [23]. Moreover, it has been demonstrated that *A. muciniphila* can be involved in anti-cancer immune therapy [24] and may prevent from atherosclerosis [25].

Cross-talk between *A. muciniphila* and the host epithelial cell in the intestine is well documented. Supplementation with *A. muciniphila* and the use of other strategies such as prebiotics or food components that increase the abundance of this bacterium in the population of gut microbiota may be a beneficial novel approach to obesity management [26]. However, the aim of this review is only to assess the effect of *A. muciniphila* supplementation versus placebo on modulating obesity.

## 2. Materials and Methods

### 2.1. Design of Primary Studies

A systematic review of clinical trials and controlled interventional studies was conducted to evaluate the role of *A. muciniphila* in obesity and lipid parameters in the obese and non-obese population.

### 2.2. Data Source and Search Strategy

All phases of implementation and reporting of this systematic review were conducted according to the Preferred Reporting Items for Systematic Reviews and Meta-Analyses Guidelines (PRISMA 2009) [27]. This systematic review has been registered with PROSPERO (19 January 2021) under number: CRD42021223143.

Studies were found by searching the following data sources: Google Scholar (https://scholar.google.com/ accessed on 31 January 2021), Web of Science (http://apps.webofknowledge.com accessed on 31 January 2021), PubMed (https://pubmed.ncbi.nlm.nih.gov/ accessed on 31 January 2021), and Medline (http://apps.webofknowledge.com/MEDLINE accessed by Web of Science on 31 January 2021) within the period January 2010–January 2021. During the search, no filters or limitations were used. Lists of references from selected studies have been scanned manually to find any other related studies.

The following search words were used, both as free-text and topic headings, in order to achieve a high-sensitivity strategy: “*Akkermansia muciniphila*”, “next-generation probiotic”, “new-generation probiotic”, “obesity”, “fat mass”, “body fat” and “lipid profile”.

The criterion to explain the “clinical trial” was based on animal experiments that may have been planned for an intervention (which may involve control or other placebo groups), and with purpose to determine the effect of ingesting *A. muciniphila* on metabolic syndrome, lipid profile, and fat mass and obesity.

### 2.3. Inclusion and Exclusion Criteria

Screening of abstracts and titles was the primary stage of the search, and the next stage consisted of checking full-text studies that met the following selection criteria: (1) animal model, (2) involvement of control group, and (3) treatment or intervention with *A. muciniphila* supplementation. The following exclusion criteria have been used: (1) not original paper, (2) lack of comparison intervention, and (3) human studies and animals with pathologies. The PICOS (Population, Intervention, Control, Outcomes, Study design) standards are shown in Table 1.

### 2.4. Data Extraction

The following data were obtained for each article: author, publication year, sample size and animal species, study design, duration and dose of ingestion of *A. muciniphila*, control and the outcomes related to obesity and metabolic syndrome.

### 2.5. Quality Appraisal

The Modified Downs and Black checklist were used to test the quality and risk of bias in the studies [28]. In 5 domains: reporting, external validity, internal validity (bias), internal validity (confounding), and power, 27 items were included in the checklist. After evaluation, the studies were classified into: excellent (between 26–28), good (between (20–25), fair (15–19) and poor (less than 14) according to their scores.

## 3. Results

### 3.1. Selection of Studies

Out of 804 articles, 262 articles were left after removal of duplicates, and 464 articles were irrelevant to the topic. Following articles screening and full text articles availability, only 78 articles have remained. Finally, after the application of inclusion and exclusion criteria, only 10 studies were considered as the main source for testing the theory of *A. muciniphila* and its association with obesity (Figure 1). After assessing the quality of the selected studies by the Modified Downs and Black checklist, 3 studies [29,30,31] showed excellent quality, and 7 studies [32,33,34,35,36,37,38] were good quality.

### 3.2. Main Results of Selected Studies

The results below were selected from the interventional studies’ outcomes as summarized in Table 2.

#### 3.2.1. *A. muciniphila* in the Improvement of Obesity Parameters

Ten studies have investigated the effect of *A. muciniphila* supplementation on the vital obesity parameters and metabolic disorder in C57BL/6J mice models [29,30,31,32,33,34,35,36,37,38]. In 2020, Yang et al. [29] found that the body weight gain, caloric intake, mesenteric fat weight, subcutaneous fat weight, epididymal fat weight, total fat and energy efficiency significantly decreased in high fat diet (HFD)-fed mice after treatment with pasteurized culture of *A. muciniphila*. Furthermore, authors examined the effects of bacterial supplementation on the colonic gene expression of Glucagon-like peptide-1 (GLP-1) and Peptide YY(PYY), which are the intestinal hormones with appetite suppressing and anti-diabetic plus anti-obesity properties. The treatment with *A. muciniphila* has increased PYY’s mRNA level, and significantly upregulated GLP-1 gene expression.

To further demonstrate whether *A. muciniphila* has to be alive to exert its metabolic effects, Everard et al. [30] have compared the effect of viable *A. muciniphila* administration with that of heat-killed *A. muciniphila* and found that viable *A. muciniphila* normalized metabolic endotoxemia caused by diet, fat storage, adipose tissue metabolism, and CD11c adipose tissue marker. Similarly, *A. muciniphila* treatment decreased body weight and improved body composition without changes of food intake. Notably, these effects were not observed after administration of heat-killed *A. muciniphila*.

Interestingly, other trials [31,32] observed that without affecting the accumulated food intake in the HFD-fed group, pasteurized *A. muciniphila* significantly decreased body weight gain, total adiposity index, and fat mass gain; additionally, the fecal caloric content significantly increased. Finally, transcript levels of the main glucose transporters GLUT2 and SGLT1 mRNA were significantly lower than in the normal diet (ND) fed group.

Depommier et al. [31] reported that five weeks of supplementation with *A. muciniphila* at a dosage of 10^8^ CFU decreased body weight gain and significantly reduced fat mass as well as increased lean mass in mice fed with a ND. Moreover, the visceral fat weight, which was more closely related to insulin resistance pathogenesis, was more clearly reduced. Furthermore, supplementation with *A. muciniphila* was unaltered in food intake and fecal triglyceride content. Shen et al. [19] showed that treatment with *A. muciniphila* maintained body weight and food intake compared to control group.

Plovier et al. [34] found that daily treatment of 2 × 10^8^ CFU of *A. muciniphila* live cells reduced HFD-induced weight gain and fat mass gain (by about 40–50%). Additionally, they observed that the same dose of pasteurized *A. muciniphila* gave a greater effect than unpasteurized culture regardless of food consumption. Furthermore, the mean adipocyte diameter is normalized, and plasma leptin is significantly lower in mice treated with pasteurized *A. muciniphila* compared to control HFD fed mice. Notably, same results were not detected in mice treated with the unpasteurized culture. The same study revealed that mice fed with pasteurized *A. muciniphila* had a higher fecal caloric content than the other groups, implying that pasteurized culture administration reduces caloric absorption. This could play a significant role in the body weight decrease and fat mass gain shown in this group. Corresponding to what was observed with the pasteurized microorganism, the treatment with Amuc_1100*, the outer membrane protein of *A. muciniphila* produced in *E. coli*, resulted in lower body weight and fat mass gain as compared to untreated HFD fed mice, regardless of food consumption. It also helped to correct the higher adipocyte diameter caused by the HFD.

Wu et al. [35] demonstrated that in both HFD and ND groups, *A. muciniphila* GP01 treatment reduced food intake and body weight. Accordingly, in the HFD group, mice ingested about 20.9 g of food daily, which was marginally higher than the amount ingested by the mice in HFD with GP01 gavage group. Comparably, the mice on the ND group ingested about 18.2 g of food per day, which was marginally more than the mice on the ND with GP01 group ingested. Moreover, after 30 weeks, the HFD mice had significantly higher body weights than the HFD-GP01 mice, and ND fed mice had a significantly higher body weight than the ND plus GP01 fed group.

Ashrafian et al. [36] reported that the HFD-mice group revealed an increase in body weight after 3 months. Obese mice were treated with *A. muciniphila*-derived extracellular vehicles (EVs) and demonstrated a substantial reduction in food consumption as well as a low level of body weight gain. As a result, obese mice feeding with *A. muciniphila* live cells caused body and epididymal adipose tissue (EAT) weight loss, but had a lower impact on body weight and adipose weight than its EVs. Notably, both interventions showed a substantial impact on body weight in ND mice, while food intake does not change. Furthermore, both the bacterium and EVs significantly reduced adipocyte size in HFD-fed mice, with the results of EVs being more observable. Finally, in comparison to the other groups, the ND group treated with *A. muciniphila* and its EVs had the smallest adipocytes.

Contradictory, Kim et al. [37] and Deng et al. [38] reported that no difference in weight gain was observed between groups treated and not non-treated group with *A. muciniphila*. Deng et al. [38] showed that the size of adipocytes in inguinal white adipose tissue (iWAT) and epididymal white adipose tissue (eWAT) was significantly increased by HFD. However, treatment with cells of strains GP01 and GP25 alleviate the effect. The diameter of inguinal adipocytes was reduced more with cells of the strain GP01, but browning was not observed in iWAT or eWAT. They also noticed that HFD mice’s scapular brown adipocytes changed from multilocular to unilocular adipocytes, a process known as brown adipose tissue (BAT) whitening. Interestingly, *A. muciniphila* treatment greatly decreased the amount of unilocular adipocytes in HFD mice, which alleviating the whitening of BAT.

#### 3.2.2. *A. muciniphila* in the Improvement of Insulin Sensitivity and Glucose Homeostasis

The results of one study [29] showed that in the HFD group, the fasting blood glucose level, determined by oral glucose tolerance test (OGTT) immediately before glucose intake (0 min), was significantly higher than in the normal fed group. However, this parameter was substantially depleted by the treatment of HFD-fed mice with *A. muciniphila*. The same authors found that in the HFD group, the OGTT area under the curve (AUC), serum insulin level, homeostatic model assessment for Insulin Resistance (HOMA-IR), and hepatic gene expression of G6Pase (an enzyme involved in glucose production) were significantly higher than in the normal group. However, the HFD group treated with *A. muciniphila* significantly reduced the levels of these four parameters. These results also showed that A. muciniphila significantly increased the hepatic GLUT2 gene expression, improving regulation of glucose transport in the liver. Moreover, this new generation probiotic significantly upregulated the expression of two important gut hormone genes, GLP-1 and PYY, involved in the appetite suppressing and anti-diabetic plus anti-obesity properties, respectively.

Everard et al. [30] demonstrated that the treatment with *A. muciniphila* completely reversed diet-induced fasting hyperglycemia and regulate the homeostasis of glucose. These effects were associated with a dramatic reduction in the expression of the hepatic glucose-6-phosphatase and the consequent drop in gluconeogenesis; insulin resistance index was also decreased after treatment. Moreover, after OGTT, viable *A. muciniphila* was found to significantly reduce plasma glucose levels, whereas heat-killed *A. muciniphila* exhibited glucose intolerance similar to that of HFD mice. On the contrary, Zhao et al. [32] found *A. muciniphila* treatment did not significantly affect the fasting blood glucose level in fed NCD mice. However, the authors reported that glucose tolerance, as reflected by the intraperitoneal glucose tolerance test (IPGTT) and the corresponding reduction of AUC, were greatly improved by *A. muciniphila* supplementation in ND fed mice. Although fasting plasma insulin levels were comparable between the two groups, a large change in liver and muscle phospho AKT Ser473 levels in the *A. muciniphila* group was detected, indicating an increased insulin sensitivity in the liver and muscles. Consequently, enhanced insulin sensitivity in the liver led to decreased hepatic G6P and Pepck expression. All these findings have demonstrated that *A. muciniphila* supplementation in ND fed mice improved energy homeostasis and glucose tolerance.

Plovier et al. [34] observed that mice given pasteurized *A. muciniphila* had significantly lower glucose intolerance and insulin concentrations than mice given the HFD, resulting in a lower insulin resistance (IR) index in the treated mice. Additionally, Amuc_1100* had the same potency as the live and pasteurized bacterium in improving glucose tolerance. The authors investigated the effects of A. muciniphila on insulin sensitivity by analyzing insulin-induced phosphorylation of the insulin receptor (IR) and its downstream mediator Akt at the threonine and serine sites in the liver. They found that HFD-fed mice got lower phosphorylation of all analyzed proteins when compared to ND-fed mice, in particular for Akt at the threonine. Interestingly, these effects were counteracted by the A. muciniphila or Amuc_1100* treatment. In particular, a significant higher phosphorylation level of IR and Akt at the threonine or Akt at the serine, when compared to untreated HFD-fed mice, was registered in mice treated with Amuc_1100* and live bacterium, respectively.

Wu et al. [35] found that mice on the HFD had significantly higher fasting blood glucose levels than mice in the other groups. Furthermore, the HFD-GP01 group’s fasting blood glucose levels were not substantially different from the ND groups. Consequently, the ND mice had significantly higher fasting blood glucose levels than the ND-GP01 fed mice. Moreover, they observe only a slight decrease in the HOMA-IR and a slight increase in the HOMA-%B in the HFD-GP01 group compared to the HFD one. The area under the oral glucose tolerance test curve showed that the HFD impaired blood glucose regulation and that GP01 significantly eased this impairment. Another study [36] showed that using EVs resulted in substantially lower plasma glucose levels in both the HFD and ND groups.

According to Deng et al. [38], *A. muciniphila* treatment reduced fasting blood glucose levels in HFD mice as compared to the HFD group. OGTT tests revealed that *A. muciniphila* intervention significantly enhanced HFD mice’s impaired glucose tolerance. They compared serum insulin levels between HFD groups and found that it was higher after *A. muciniphila* gavage, implying that the microorganisms might be able to stimulate insulin release to lower blood glucose levels. However, only the treatment with the strain GP01 was found to be significantly involved in these results.

#### 3.2.3. *A. muciniphila* in Improvement of Anti-Inflammatory Effects

In addition to the role of *A. muciniphila* on obesity and glucose metabolism, it was thought to be involved in inflammation modulation. The results of Yang et al. [29] showed that in the HFD group, TNF-a, IL-6, MCP-1, TLR2 and TLR4 colonic mRNA levels were significantly higher than in the normal group. Conversely, the colonic expression of the IL-10 gene was significantly lower. The HFD group’s exposure to *A. muciniphila* has substantially depleted the TLR2 mRNA level and downregulated the expression of the TNF-a and MCP-1 genes.

Zhao et al. [32] found that the supplementation with *A. muciniphila* significantly decreased the phospho-JNK level and increased the level of the NF-KB protein inhibitor and IKBA protein in the liver, indicating that inactivation of these two pathways in the supplementation of the *A. muciniphila* group has enhanced metabolic endotoxemia and subsequent local inflammatory cascades, which could mediate the beneficial metabolic effects.

Plovier et al. [34] reported that HFD-fed mice had higher portal LPS levels than ND-fed mice, but intervention with *A. muciniphila* (live or pasteurized) or Amuc_1100* protein completely restored LPS levels to those of ND group.

Wu et al. [35] found that the mice in the ND-GP01 group had significantly higher IL-10 levels than the mice in the other groups. The HFD-GP01 group’s IL-10 levels were significantly higher than those of the HFD and ND groups. This result showed that intervention with *A. muciniphila* GP01 improved the expression of the anti-inflammatory factor IL-10.

Ashrafian et al. [36] showed that the treatment with *A. muciniphila* decreased the mRNA expression of TLR-4 and IL-6 genes in EAT in both HFD and ND fed mice, but had no effect on TNF-α expression in HFD mice, whereas it increased in ND mice. In HFD mice, however, EVs caused a greater reduction in inflammatory cytokines (TNF-a and IL-6) and TLR-4 expression. However, when compared to the bacterium, EVs reduced more TNF-a and TLR-4 expression in ND mice.

In HFD mice, Deng et al. [38] found that *A. muciniphila* GP01 administration significantly reduced the mRNA expression of macrophage inflammation markers (Itgax and Emr1), immune cell recruitment (Ccl2), and LPS-binding protein (Lbp), while *A. muciniphila* GP25 gavage only significantly reduced *Emr1* and *Lbp* transcript expression. Furthermore, in ND mice, the strain GP01 significantly decreased Lbp mRNA expression. These findings suggested that *A. muciniphila* gavage could reduce inflammation in BAT and that GP01 intervention improved gene regulation.

#### 3.2.4. *A. muciniphila* in Lipid Levels and Metabolism

Oral gavage of *A. muciniphila* in three studies has shown that plasma triglycerides levels were unaltered [31,32,33,37]. Shen et al. [33] also found that there was no significant change in the cholesterol contents after treatment.

Another study [34] showed that *A. muciniphila* administration or its protein extract substantially reduced plasma HDL cholesterol concentrations and reversed HFD-induced hypercholesterolemia. Pasteurized *A. muciniphila* also showed lower plasma triglyceride levels than both untreated mice or HFD-fed mice treated with live *A. muciniphila*.

Ashrafian et al. [36] showed that the EVs corrected HFD-induced hypercholesterolemia with significantly lower plasma TC, but TG levels did not change. On the other hand, obese mice and normal group treated with *A. muciniphila* reported significantly lower plasma TG.

Deng et al. [38] demonstrated that the *A. muciniphila* GP01 intervention improved both TG and TC, while *A. muciniphila* GP25 only improved TC. When TC levels between the HFD groups were compared, GP25 showed a more substantial difference.

## 4. Discussion

One of the most common causes of health issues is obesity, which can contribute to the development of other severe metabolic diseases [39]. Clinical trials have been performed with pharmacological agents and other therapeutic interventions, including surgery, but there are currently no effective, specific obesity therapeutics available [40]. Recently, steps towards the growing use of *A. muciniphila* as a next-generation probiotic have been successfully achieved. First, the observation that *A. muciniphila* is a cultivable microorganism is consistent with human implementation. Second, the observation that the pasteurization of the bacteria enhanced its impact, and therefore its durability and possible lifespan. Third, identifying main interaction mechanisms between *A. muciniphila* and its host. Fourth, the manifestation of the safe administration of *A. muciniphila* in the individual targeted population [41].

It is recognized that *A. muciniphila* could play a crucial role on the obesity parameters in human and different mice models, such as reduced body weight, fat mass, hip circumference, caloric intake, mesenteric fat weight, subcutaneous fat weight, epididymal fat weight, total fat and energy efficiency [29,30,31,32,33,34,35,36,37,38].

The treatment of HFD fed animals with *A. muciniphila* increased PYY’s mRNA level, and significantly upregulated GLP-1 gene expression, which suppress the appetite and control feeding [29,30]. As evidence, Lee et al. [42] reported that GLP-1 promotes secretion of insulin, inhibits secretion of glucagon and suppresses appetite and food intake thanks to the increased satiety and slow gastric emptying. On the other hand, PYY was shown to reduce surplus adiposity, enhance glucose tolerance, and modulate body weight by inhibiting the appetite and increasing energy expenditure [43,44]. As matter of fact, some authors [34,36,37,38] observed that *A. muciniphila* administration, like active or pasteurized culture, reduced adiposity size (e.g., iWAT and eWAT) in HFD fed mice. It could be hypothesized that, in these cases, the effect is related to the expression of the gene involved in the appetite suppression.

*A. muciniphila* supplementation improved glucose tolerance and insulin sensitivity in C57BL/6 mice fed with HFD [29,30,31,32], suggesting that a diet reinforced with this microorganism could bring benefits to a metabolic healthy subject because of an overall decrease in fat mass and increase in lean mass. However, Everard et al. [30] did not find similar changes in ND fed C57BL/6 mice. The discrepancy might be due to the difference in the length of treatment or to the diet content. As a matter of fact, Zhao et al. [32] and Everard et al. [30] treated mice with *A. muciniphila* for 5 and 4 weeks, respectively. Furthermore, they applied a normal diet with a different proportion of fat, fiber, proteins and carbohydrates, suggesting that the effects may be diet or time dependent. Moreover, Deng et al. (38) found that the strain GP01 increased insulin level in HFD mice, but the authors reported a significant decrease in blood glucose level with both GP01 and GP025 strains, compared to HFD control. This result could suggest that the effect of *A. muciniphila* on glucose level is not related to the insulin level, but to the insulin sensitivity.

Low-grade inflammation conditions, such as obesity and type 2 diabetes, are affected by gut microbiota, which increases intestinal permeability through the exposure of tight junction proteins to lipopolysaccharide (LPS) [45]. In this scenario it was demonstrated that A. muciniphila, due to its gut barrier maintenance ability, significantly reduced plasma LBP levels [31], suggesting a reduced metabolic endotoxemia in ND fed mice, because LBP is an indicator of circulating LPS [30]. Moreover, it was demonstrated that adipose tissue differentiation and lipogenesis are inhibited by higher circulating LPS levels [30,34], so contributing to altered adipose tissue metabolism typical of obesity condition [46]. These results could demonstrate the improved endotoxemia and chronic inflammation by *A. muciniphila* supplementation.

It was stated that *A. muciniphila* can increase anti-inflammatory cytokine levels and decrease pro-inflammatory conditions to improve systemic inflammation in mice caused by HFD [29,32,34,35,36,38]. Consistent with these findings, some authors [29,35] indicated that intervention of the HFD group mice with different *A. muciniphila* strains significantly increased the colonic gene expression of IL-10 and decreased the colonic mRNA levels of TNF-a, MCP-1 and TLR2. In comparison, treatment of HFD-fed animals with EB-AMDK 19 and EB-AMDK 27, which are considered *A. muciniphila* strains, substantially decreased the expression of IL-6 and TLR4 genes in the gut [29,36]. However, Ashrafian et al. [36] found that EVs induced more reduction in the inflammatory cytokines TNF-α and IL-6 and in the expression of TLR-4 in HFD mice. Instead, ND-EVs fed mice showed a higher reduction in TNF-α and TLR-4 expression compared to the microorganism.

## 5. Conclusions

This review strongly suggests that supplementation with *A. muciniphila* is an effective new-generation beneficial bacterium in decreasing obesity parameters, improving insulin sensitivity and glucose homeostasis, modulating energy homeostasis and improving inflammation. However, there is a need for further investigation on different *A. muciniphila* strains, especially on humans.

## Figures and Tables

**Figure 1 microorganisms-09-01098-f001:**
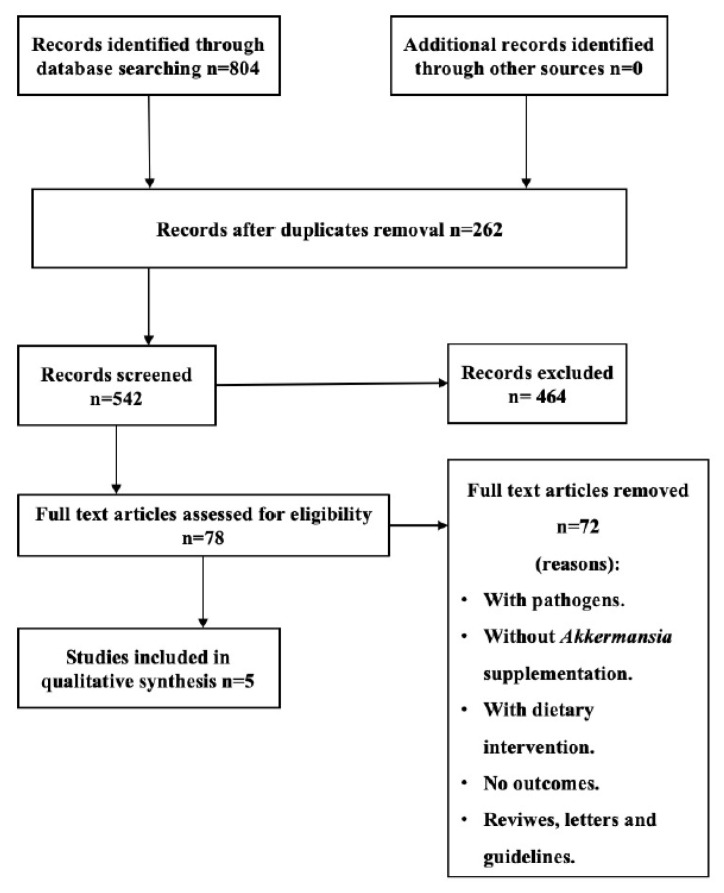
PRISMA flowchart showing the progression of trials through each stage of the selection process.

**Table 1 microorganisms-09-01098-t001:** Criteria of PICOS standards for articles selection.

Standards	Inclusion Criteria	Exclusion Criteria
Population	Mice models.	Human, other animal models and mice with pathology.
Intervention	Oral administration of *A. muciniphila* (alive cells, eat treated culture or cell extracts) combined to pre-treatment with normal diet or high fat diet.	No administration of *A. muciniphila* or increase of *A. muciniphila* population in the microbiota related to diet.
Control	Placebo, vehicle.	No control.
Outcomes	Lipid metabolism and inflammation markers, circulating parameters, metabolic parameters, body weight, calories intake, adipocytes and fat mass.	Poor procedure or no clear findings, other outcomes.
Study design	Controlled studies with a separate control group.	Case studies, cross-over studies, studies without a separate control group.

**Table 2 microorganisms-09-01098-t002:** Effects of *Akkermansia muciniphila* supplementation on obesity and metabolic parameters of male C57BL/6 mice.

Reference	Intervention	Control	Dosage and Duration	Outcomes
[29]	- *A. muciniphila* BAA 835^T^ with HFD * - *A. muciniphila* EB-AMDK 10 with HFD - *A. muciniphila* EB-AMDK 19 with HFD - *A. muciniphila* EB-AMDK 27 with HFD	LFD * HFD	1 × 10^8^ CFU 12 weeks	- decreased body weight gain - decreased caloric intake - reduction in the weights of the major adipose tissues and total fat - improved glucose homeostasis and insulin sensitivity - inhibition of low-grade intestinal inflammation
[30]	- oral gavage of *A. muciniphila* BAA835^T^ with ND * - oral gavage of *A. muciniphila* BAA835^T^ with HFD	ND HFD	2 × 10^8^ CFU 4 weeks	- improved metabolic profile - decreased fat-mass gain and metabolic endotoxemia - decreased adipose tissue inflammation - decreased insulin resistance
[31]	- oral gavage of pasteurized *A. muciniphila* BAA-835^T^ with HFD including pre-treatment with HFD - oral gavage of pasteurized *A. muciniphila* BAA-835^T^ with ND including pre-treatment with HFD	ND HFD	2 × 10^8^ CFU 5 weeks	- reduced body weight gain and fat mass gain - enhanced energy expenditure and oxygen consumption - enhanced physical activity - increases fecal energy content - reduces expression of carbohydrates transporters - influences expression of lipid-droplet regulator
[32]	- oral gavage of *A. muciniphila* BAA-835^T^ with ND including pre-treatment with ND	ND	2 × 10^8^ CFU 5 weeks	- decreased body weight gain - reduced fat mass - improved glucose tolerance and insulin sensitivity - reduced gene expression related to fatty acid synthesis and transport in liver and muscle - reduced chronic low-grade inflammation - decreased plasma levels of lipopolysaccharide - increase in anti-inflammatory factors
[33]	- oral gavage of *A. muciniphila* BAA-835^T^ with ND including pre-treatment with ND	ND	2 × 10^8^ CFU 2 weeks	- maintain body weight - maintain food intake - no significant changes in the plasma triglyceride and cholesterol contents
[34]	- oral gavage of pasteurized *A. muciniphila* with HFD - oral gavage of alive *A. muciniphila* with HFD - *A. muciniphila* protein Amuc_1100 * with HFD	ND HFD	2 × 10^8^ CFU 3 µg protein 4 weeks	- decreased body weight - reduced fat mass - improved glucose homeostasis and insulin sensitivity - inhibition of low-grade intestinal inflammation - lower adipocyte diameter
[35]	- oral gavage of *A. muciniphila* GP01 with ND - oral gavage of *A. muciniphila* GP01 with HFD	ND HFD	1 × 10^9^ CFU 10 months	- reduced body weight - decreased food consumption - improved blood glucose - improved inflammation
[36]	- oral gavage of *A. muciniphila* with HFD including pre-treatment with HFD - oral gavage of *A. muciniphila* with ND including pre-treatment with ND - oral gavage of extracellular vesicles with ND including pre-treatment with ND - oral gavage of extracellular vesicles with HFD including pre-treatment with HFD	ND HFD	1 × 10^9^ CFU 10 μg protein 5 weeks	- reduced body weight gain and adipose weight gain - reduced food intake - lower plasma TG - lower plasma glucose - reduced gut permeability and upregulate tight junction - improved energy metabolism - induced pro- and anti-inflammatory cytokine secretion
[37]	- oral gavage of *A. muciniphila* with HFD - oral gavage of *A. muciniphila* with ND	ND HFD	1 × 10^9^ CFU 10 weeks	- no difference in body weight gain - lower TG - maintain gut homeostasis
[38]	- oral gavage of *A. muciniphila* GP01 with ND - oral gavage of *A. muciniphila* GP01 with HFD - oral gavage of *A. muciniphila* GP25 with HFD - oral gavage of *A. muciniphila* GP25 with ND	ND HFD	5 × 10^9^ CFU 16 weeks	- improved glucose homeostasis - improved lipid metabolism - reduced serum TG and TC - diminished adipocyte size in WAT and BAT - alleviated inflammation - improved endotoxemia

* HFT, high fat diet; LFD, low fat diet; ND, normal diet.

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
