# Peer review of "Akkermansia muciniphila, a New Generation of Beneficial Microbiota in Modulating Obesity: A Systematic Review"

_microorganisms, 2021, doi:10.3390/microorganisms9051098_

Round 1

Reviewer 1 Report

The topic is interesting and relevant, the manuscript is generally well-written. However, there are some points which should be improved prior to publication. I would recommend a more extensive description of the 'A. muciniphila in the improvement of insulin sensitivity and glucose homeostasis'.

(page 7, 10~14-th lines) In the reference [38], it was estimated that the serum insulin levels (ng/mL) was measured in fasting state. Despite the HFD model, GP01 treatment in HFD model showed very high serum insulin levels compared to control (HFD+PBS). The hyper secretion of insulin in the fasting state compared to control was thought to hyperinsulinemia. It was difficult to improve the glucose homeostasis and induce balanced glucose homeostasis. In the reference [29], the serum insulin levels in fasting state was significantly reduced by A. muciniphila treatment compared to control (HFD model). That is, the insulin secretion was safely controlled in fasting state, suggesting that the attenuated insulin secretion can be contributed to improvement of glucose homeostasis. Although the contradictory data was showed in the references [29, 38], the consideration was not described in this manuscript.

(page 9, 20-th line) 'A. muciniphila' should be italicized.

Author Response

We extended the section 'A. muciniphila in the improvement of insulin sensitivity and glucose homeostasis' according to the reviewer' suggestion.

In our opinion the results of Deng et al. (2020) and Yang et al. (2020) are just apparently contrasting. In fact, even though different levels of insulin, both found an improvement of glucose homeostasis. We added a sentence taht could clarifiy the point.

Reviewer 2 Report

The manuscript entitled « Akkermansia muciniphila, a new generation of beneficial microbiota in modulating obesity: A systematic review” reviews the recent discoveries of the effects of Akkermansia muciniphila on obesity and obesity-related disorders (glucose intolerance, insulin resistance, inflammation). The topic is timely, the text is comprehensive and the overall impression is positive. However, some references are missing and are not up to date. Moreover, some sections need to be better explained.

Considering that all the review is based on the effects of Akkermansia muciniphila on obesity, it appears relevant to describe how the abundance of Akkermansia muciniphila is modulated. Authors wrote “the use of other strategies such as prebiotics or food components […] obesity management” but this section should be further developed. Authors need to cite some relevant and recent papers that described how prebiotics, nutritional compounds, polyphenols, fibers etc increase the population of Akk. muciniphila and the effects of HFD and other diets on this bacterium.

In the introduction, authors need to be more precocious when they talk about the ratio Firmicutes to Bacteroides. This is controversial and some studies have demonstrated an opposite effect. Please, put some balance in your statement.

Still in the introduction, authors should modify this sentence that is not in agreement with the rest of their remarks “The presence of A. muciniphila was correlated with weight gain in human and animal models”. Please, check carefully the reference.

The rest of the review reads much better but some details are missing to really understand the purpose of the review.

When authors explain the effects of Akkermansia supplementation on obesity and obesity-related disorders, they should mention more accurately the type of diet that has been used in each study. Quantity of fibers in control diet varies and can contribute to the effects observed. Moreover, the quantity of fat and sugars is different among the different HFD.

Since the effects of Akkermansia supplementation on food intake seem controversial, authors are encouraged to detail the way authors are measuring it (metabolic cages, weekly measure…).

Please, write the dose of Akkermansia that is used in each study.

Authors mainly focused on animal studies. Over the past years, the first effects of Akkermansia in humans have been described. Please, comment these studies.

It has been described that HFD and HFHS diets do not have the same effects in the modulation of Akkermansia. Please add a sentence and references on this point in the discussion to balance your purpose.

Author Response

Q. Considering that all the review is based on the effects of Akkermansia muciniphila on obesity, it appears relevant to describe how the abundance of Akkermansia muciniphila is modulated. Authors wrote “the use of other strategies such as prebiotics or food components […] obesity management” but this section should be further developed. Authors need to cite some relevant and recent papers that described how prebiotics, nutritional compounds, polyphenols, fibers etc increase the population of Akk. muciniphila and the effects of HFD and other diets on this bacterium.

R. The modulation of "autochthonous probiotics" by the diet is an enormous topic that we cannot be taken in consideration in this review, which is focused just on the diet supplementation with A. muciniphila.

Q. In the introduction, authors need to be more precocious when they talk about the ratio Firmicutes to Bacteroides. This is controversial and some studies have demonstrated an opposite effect. Please, put some balance in your statement.

R. We changed accordingly.

Q. Still in the introduction, authors should modify this sentence that is not in agreement with the rest of their remarks “The presence of A. muciniphila was correlated with weight gain in human and animal models”. Please, check carefully the reference.

R. We deleted the sentence.

Q. When authors explain the effects of Akkermansia supplementation on obesity and obesity-related disorders, they should mention more accurately the type of diet that has been used in each study. Quantity of fibers in control diet varies and can contribute to the effects observed. Moreover, the quantity of fat and sugars is different among the different HFD. Since the effects of Akkermansia supplementation on food intake seem controversial, authors are encouraged to detail the way authors are measuring it (metabolic cages, weekly measure…).

R. We underlined in the Discussion section the weakness in the comparison of these types of experiments in which there is not standardization of protocols. However, I think the reviewer agree with us that this is the same problem for many other fields of research, but when many results are taken into account, the effects related to some differences in the protocols could be considered null.

Q. Please, write the dose of Akkermansia that is used in each study.

R. Dosage is reported in the table 2.

Q. Authors mainly focused on animal studies. Over the past years, the first effects of Akkermansia in humans have been described. Please, comment these studies.

R. We decided to write a systematic review just to focus on the papers responding to the inclusion criteria, and the first criterion is "animal model".

Q. It has been described that HFD and HFHS diets do not have the same effects in the modulation of Akkermansia. Please add a sentence and references on this point in the discussion to balance your purpose.

R. We think that this is an important point but out of topic of this review. As previously reported we didn't mention papers in which the modulation of Akkermansia is investigated but just its supplementation.